# A signal transmission strategy driven by gap-regulated exonuclease hydrolysis for hierarchical molecular networks
Xin Liu [1], Xun Zhang [1], Shuang Cui [1], Shujuan Xu[2], Rongming Liu [3], Bin Wang [4], Xiaopeng Wei[1] & Qiang Zhang [1] ✉

Exonucleases serve as efficient tools for signal processing and play an important role in biochemical reactions. Here, we identify the mechanism of cooperative exonuclease hydrolysis, offering a method to regulate the cooperative hydrolysis driven by exonucleases through the modulation of the number of bases in gap region. A signal transmission strategy capable of producing amplified orthogonal DNA signal is proposed to resolve the polarity of signals and byproducts, which provides a solution to overcome the signal attenuation. The gap-regulated mechanism combined with DNA strand displacement (DSD) reduces the unpredictable secondary structures, allowing for the coexistence of similar structures in hierarchical molecular networks. For the application of the strategy, a molecular computing model is constructed to solve the maximum weight clique problems (MWCP). This work enhances for our knowledge of these important enzymes and promises application prospects in molecular computing, signal detection, and nanomachines.

In biological systems, signal transmission and transduction are required to regulate the complex biochemical processes that are indispensable in biochemical reaction networks. The types of signals encompass both physical and chemical signals[1], and the mode of their transmission directly determines the characteristics of the entire system. In addition, the programmability of the signal path is crucial for complex signal processing and the construction of large-scale hierarchical biochemical networks[2,3], which imposes challenges on the compatibility and universality of signal processing in biochemical systems.

For the processing of biochemical signals, numerous in vitro transmission strategies for DNA signals have been developed and applied to complex biochemical reaction systems. For example, DNAzyme-based systems have the capability to process the signal of RNA-modified DNA[4,5]. The strategy of optochemical control has been used as an external trigger for biological regulation[6,7]. DNA primers trigger signal output through the combination of polymerase and nickase enzymes[8,9]. Double-stranded DNA (dsDNA) is hydrolyzed by the exonuclease to output the signal of single-stranded DNA (ssDNA)[10,11]. The pH-dependent regulation of the Hoogsteen hydrogen bond alters DNA structure[12,13]. The binding of DNA aptamer with its target molecule has been developed as a strategy for signal

transmission[14,15]. These DNA reaction systems provide excellent solutions for signal transmission and used to be combined with DSD mechanism[16,17].

DSD is a well-established signal transmission strategy that has been widely applied in the fields of molecular computing[18,19], nanomachines[20,21], and signal detection[22,23]. The powerful function of DSD is due to its excellent signal exchange characteristics, controllable reaction rate, strong regulatory abilities, and high compatibility, making it applicable to most DNA reaction systems[16]. In a typical DSD reaction, the 3' (5') end of the input ssDNA displace the 5' (3') end of another DNA while strictly following the directionality of the input DNA. As the displaced strand separates, the invading DNA forms new inactive dsDNA waste in solution. However, biochemical reactions are typically incomplete and accompanied by signal loss, resulting in an input-to-output ratio greater than 1 in practical reactions[24]. Achieving a higher input-output ratio can be realized by using a catalytic DSD reaction[25], which increases the complexity of the system. In addition, the waste accumulation and signal loss in DSD processes may pose significant challenges for the transmission or branching of signals over long distances, and the construction of large-scale hierarchical biochemical networks.

Exonucleases serve as efficient tools for signal processing and play an important role in biochemical reactions such as DNA replication, repair,

[1]School of Computer Science and Technology, Dalian University of Technology, Dalian 116024 Liaoning, China. [2]Key Lab of Biotechnology and Bioresources Utilization of Ministry of Education, College of Life Science, Dalian Minzu University, Dalian 116600 Liaoning, China. [3]MOE Key Laboratory of Bio-Intelligent Manufacturing, School of Bioengineering, Dalian University of Technology, Dalian 116024 Liaoning, China. [4]Key Laboratory of Advanced Design and Intelligent Computing, Ministry of Education, School of Software Engineering, Dalian University, Dalian 116622 Liaoning, China. ✉e-mail: zhangq@dlut.edu.cn

and recombination[26–28]. Exonucleases hydrolyze dsDNA from its 3' or 5' ends, and the hydrolysis properties can be utilized for signal amplification, recognition, and transmission[29–31], thus enabling the elimination of dsDNA waste and the amplification of signals. Taking exonuclease III (Exo III) and exonuclease λ (Exo λ) as examples, Exo III (λ) hydrolyzes DNA from the 3' (5') end of dsDNA[32,33]. The hydrolysis substrates for Exo λ need to be modified with phosphate or FAM at the 5' hydrolysis initiation site or follow the rules of base mismatch[34,35], whereas Exo III needs no modifications. Unlike most enzymes such as APE1 and APE2 that have evolved into endonuclease and exonuclease separately, Exo III, as a multifunctional enzyme, possesses both exonuclease activity and endonuclease activity at the apurinic/apyrimidinic (AP) site[36]. dsDNA with an AP site can be hydrolyzed by the endonuclease activity of Exo III[27], and then, Exo III continues to hydrolyze along the 3' end of the cleaved domain through the exonuclease activity while the 5' end of the cleaved AP site would be endowed with a phosphorylate that cannot be hydrolyzed by Exo III[37]. The remaining 5' recessed end serves as a substrate for Exo λ[38], allowing complete hydrolysis of the dsDNA containing the AP site by the combination of Exo III and Exo λ, which provides a feasible for dealing with the waste accumulation and signal loss. However, the DNA gap region that is opposite to the AP site may influence the hydrolysis of Exo III and Exo λ; this will determine whether the region for the endonuclease activity of Exo III must be dsDNA, thus offering the method to control the cooperative hydrolysis driven by Exo III and Exo λ through the bases in the gap region.

Here, we disclose the mechanism that the length of the DNA gap region opposing the AP site on the dsDNA regulates the hydrolysis process of the enzymes. Modulating the number of bases in the DNA gap effectively triggers or inhibits the hydrolytic activity of Exo III. However, only the products cleaved by the endonuclease activity of Exo III can subsequently be hydrolyzed by Exo III (3'-5') and Exo λ (5'-3') from different directions on the cleaved dsDNA strands, thereby consuming the entire DNA containing the AP site to release two DNA strands constituting the gap. Based on this mechanism, we propose a signal transmission strategy in which the input signal catalyzes the generation of output signals with orthogonal characteristics. This strategy not only resolves the polarity of directional DSD but also eliminates byproducts, resulting in a solution containing only the input and output signals. Thus, the approach is theoretically provided to overcome the signal attenuation during long-distance transmission and demonstrates the capability of amplifying weak signals. Moreover, the gap-regulated mechanism combined with DSD reduces the unpredictable secondary structures, allowing the coexistence of similar structures in a system. To demonstrate the compatibility, universality, and applicability of the strategy for use in hierarchical molecular networks, we construct a

molecular computing model consisting of a Point module, Weight module, and Sum & Output module and develop a corresponding algorithm to solve NP-hard problems such as the MWCP, which is of significance in protein structure prediction, computer vision, and robotics[39]. We anticipate that the gap-regulated hydrolysis mechanism and strategy can be widely applied in signal detection, molecular computing, and nanomachines.

## Results

### Cooperative hydrolysis driven by Exo III and Exo λ

As a multifunctional enzyme, Exo III possesses exonuclease activity toward blunt-ended and recessed-ended dsDNA, as well as endonuclease activity toward dsDNA containing an AP site. The cooperative hydrolysis driven by Exo III and Exo λ on dsDNA containing an AP site is illustrated in Fig. 1a. To prevent Exo III-mediated hydrolysis of the 3' end of AB, four thymine bases with phosphorothioate modifications were incorporated at the 3' ends of A and B (black tails in Fig. 1a)[22]. The AP site undergoes electrostatic adsorption by Exo III, resulting in the rapid creation of a gap on AB, and Exo III continuously hydrolyzes the 3' end of the gap due to its exonuclease activity. The hydrolysis process completely consumes A1 and converts AB into A2B with a 5' recessed end, which cannot be further hydrolyzed by Exo III (curve 2 in Fig. 1b, c, lane 4 in Fig. 1d). As Exo III generates a phosphorylated 5' recessed end, which serves as the substrate for Exo λ hydrolysis, the remaining product A2B, can be further hydrolyzed by Exo λ, ultimately producing ssDNA B and several sequence fragments containing phosphorothioate modifications released from the unstable dsDNA structure (lane 7 in Fig. 1d). However, the FAM fluorescence intensity at 60 min is higher for AB + F1 + F2 + III (curve 2) than for AB + F1 + F2 + III + λ (curve 4) in Fig. 1b. The difference in fluorescence intensity is that the addition of Exo λ promotes the generation of complete B, which simultaneously opens F1 and F2. The opened F2 with BHQ2 has a certain quenching effect on FAM fluorescence (Supplementary Fig. 1c). The inability of Exo λ to hydrolyze unmodified A2B (lane 8 in Fig. 1d) confirms the essential prerequisite of Exo III endonuclease hydrolysis for initiating the hydrolysis of Exo λ. However, once F1 is opened by A2B, which has not yet undergone hydrolysis, the subsequent hydrolysis of A2B by Exo λ can be inhibited (curves 4 and 5 in Fig. 1c). The inhibitory effect has been discovered and applied to a signal transmission strategy[10]. Therefore, to ensure complete hydrolysis of A, it is imperative to abstain from the addition of F1 in the solution. Instead, the addition of exclusively F2 is required for the analysis and detection of the output (curve 5 in Fig. 1c).

Figure 1e compares the hydrolytic reactions of dsDNA with unphosphorylated AP site (A2(AP)B), phosphorylated base (A2(PO₄)B), and unmodified dsDNA (A2B). It can be observed that A2(PO₄)B exhibits the

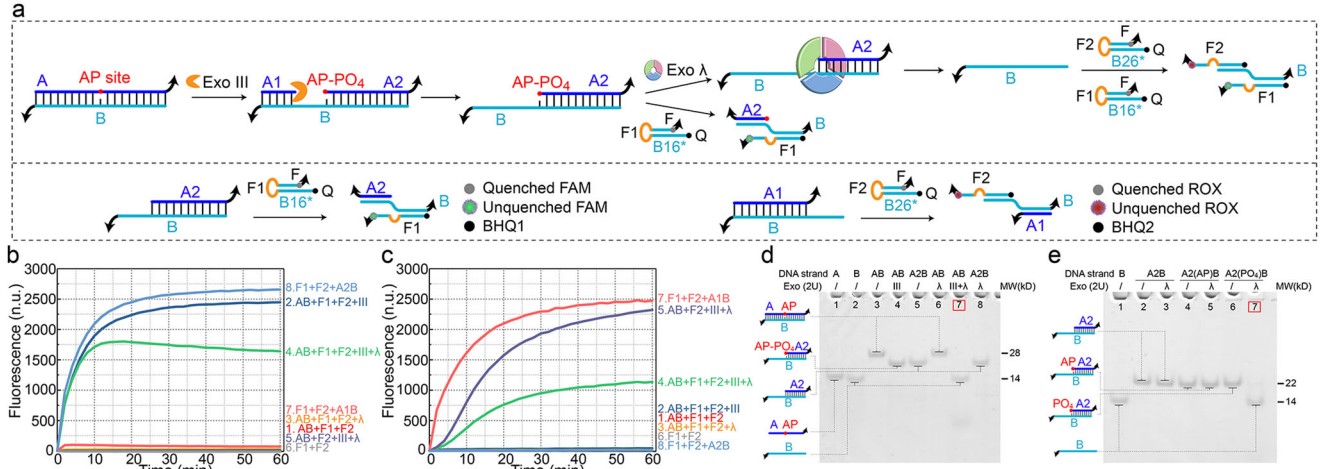

**Fig. 1 | Cooperative hydrolysis driven by exonucleases on the dsDNA with an AP site. a** Schematic illustration of the hydrolysis process. **b** FAM fluorescence testing during the reaction process. **c** ROX fluorescence testing during the reaction process. **d** PAGE verification of the hydrolysis of dsDNA with an AP site. **e** PAGE verification of the hydrolysis of 5' phosphorylated dsDNA.

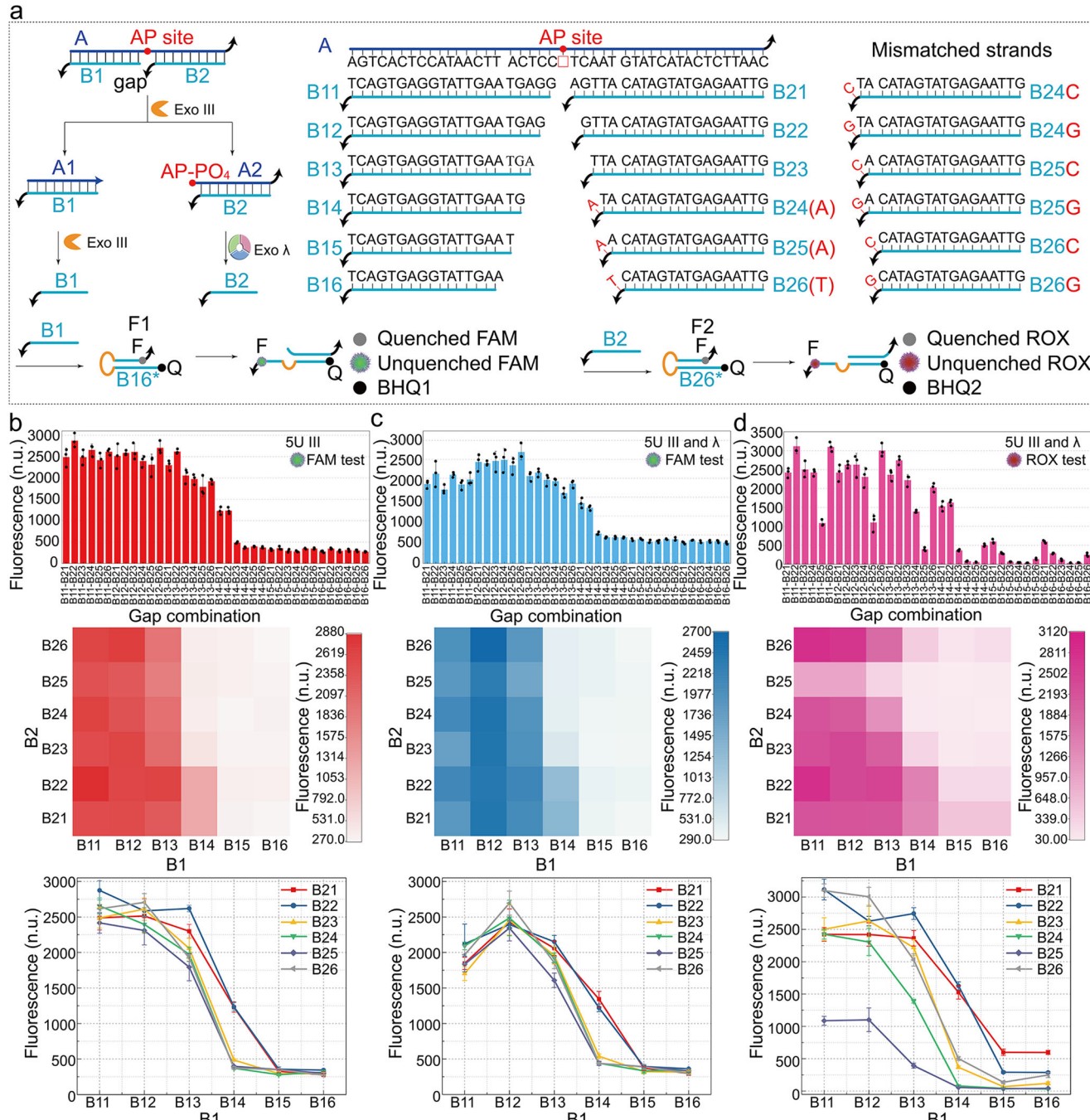

**Fig. 2 | Mechanism of gap-regulated exonuclease hydrolysis. a** Schematic illustration of gap-regulated exonuclease hydrolysis. **b** Fluorescence of substrate hydrolysis with gap combinations at the background of 5U Exo III by FAM. **c** Fluorescence of substrate hydrolysis with gap combinations at the background of 5U Exo III and Exo λ by FAM. **d** Fluorescence of substrate hydrolysis with gap combinations at the background of 5U Exo III and Exo λ by ROX. The error bars represent the standard deviation of three sets of data ($n = 3$ independent experiments).

fast hydrolysis rate (lane 7 in Fig. 1e), whereas A2B or A2(AP)B cannot be hydrolyzed by Exo λ (lanes 3 and 5 in Fig. 1e), which confirms that the phosphorylated bases generated by the endonuclease activity of Exo III at the AP site are crucial for triggering subsequent exonuclease hydrolysis of Exo λ.

## Mechanism of gap-regulated exonuclease hydrolysis

Based on the aforementioned characteristics of Exo III and Exo λ, it is reasonable to suppose that the gap region opposite to the AP site on the dsDNA has the potential to control the endonuclease hydrolysis of Exo III. Therefore, we explored the impact of varying the number of bases at the gap

region on the cooperative hydrolysis by adjusting the lengths of two ssDNA strands (B1 and B2) complementary to the ssDNA (A) containing the AP site, as shown in Fig. 2a. The distance from the complementary region to the AP site for B1 and B2 is 1 to 6 bases, resulting in a total of 6 × 6 combinations. Mismatched bases were introduced in B24–B26 to impede the hybridization between the black tail (phosphorothioate modified T bases) and A. The substrate AB1B2 with the gap region ultimately produces B1 and B2 under the cooperative hydrolysis driven by Exo III and Exo λ. Through hydrolysis tests of substrates with all gap combinations (Supplementary Figs. 2–5), we found that the presence of Exo III results in the generation of FAM signals (Fig. 2b, c), whereas the generation of ROX signals occurs only when both

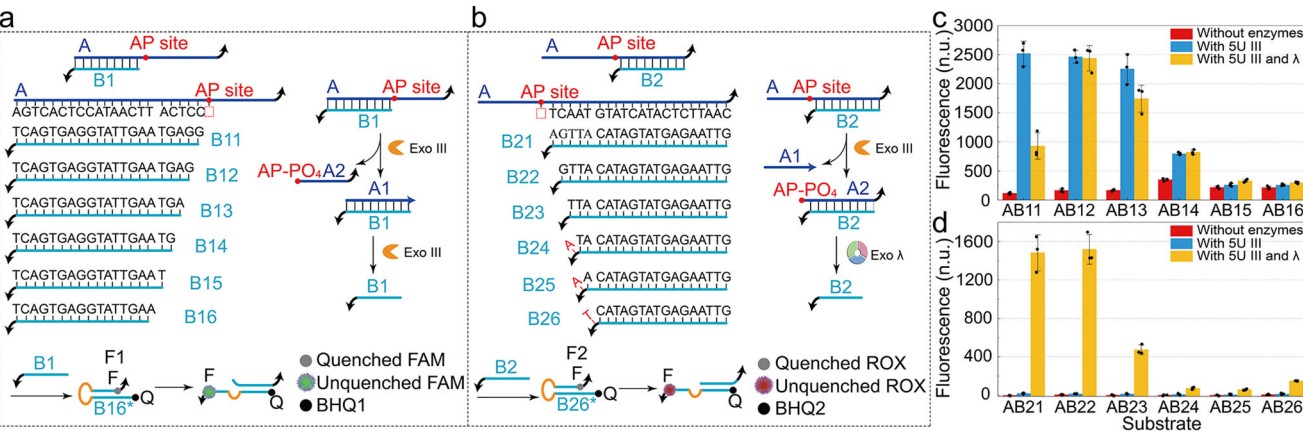

**Fig. 3 | Hydrolysis of partial duplex exposed at AP site. a** The hydrolysis of partially exposed dsDNA covered by B1. **b** The hydrolysis of partially exposed dsDNA covered by B2. **c** FAM measurement for the hydrolysis of AB1. **d** ROX measurement for the hydrolysis of AB2. The error bars represent the standard deviation of three sets of data ($n = 3$ independent experiments).

Exo III and Exo λ are present (Fig. 2d). The hydrolysis of substrates is adjusted by changes in the length of the gap region. The change in the base number of B1, which is a more significant factor compared to B2, demonstrates good monotonicity on the hydrolysis efficiency at the constant length of B2 (heatmaps and curves in Fig. 2b–d). In addition, since the endonuclease hydrolysis of Exo III is crucial to initiate the hydrolysis of Exo λ, the product of B2 (Fig. 2d) is expected to remain consistent with the product of B1 in Fig. 2b. The PAGE gel analysis of the substrates with all gap combinations is consistent with the fluorescence results (Supplementary Fig. 6).

There are additional interesting results from the hydrolysis experiments depicted in Fig. 2a. The fluorescence attenuation with the addition of Exo III and Exo λ was observed compared to the addition of Exo III (columns B11 and B13 in Supplementary Fig. 2). F1 and F2 perform excellent resistance to both enzymes in Supplementary Fig. 7. In addition, the 5' end of B1 with a dangling tail in the opened F1 (Fig. 2a) is a potential substrate for Exo λ hydrolysis[35]. Therefore, the impact of different enzymes added during the opening of the hairpin F1 on fluorescence signals was tested. We found that the addition of Exo λ is the primary cause of signal attenuation, particularly in the cases of B11 and B13 (Supplementary Fig. 8). Moreover, the continuous reduction in the length of the 5' end of B1 promotes the opened F1 to close, resulting in a decrease in fluorescence signal. In contrast, the opened F2 with a 5' recessed end of B2 causes a fluorescence increase with the addition of Exo λ (blue curves in Supplementary Fig. 9). Furthermore, hydrolysis rate of Exo λ dramatically decreases for the gap formed by B25 with B11 to B16 (row B25 of the heatmap in Fig. 2d and Supplementary Fig. 3), whereas the endonuclease effect of Exo III on the substrate is not limited (lane 5 of III, VI, IX, and XII in Supplementary Fig. 6). Exo λ hydrolyzes the product (A2(AP-PO$_4$)B2 in Fig. 2a) cleaved from Exo III at a low rate (lane 6 of III, VI, IX, and XII in Supplementary Fig. 6). The hydrolysis of A2B2 with 5' phosphorylated end is tested in Supplementary Fig. 10a to compare with A2(AP-PO$_4$)B2. The complete hydrolysis of A2B2 with 5' phosphorylated end under different lengths of B2 was achieved within 1 h or even 0.5 h (Supplementary Fig. 10b, c), which is opposite to the result that A2(AP-PO$_4$)B2 cleaved from Exo III cannot be further hydrolyzed by Exo λ within 1 h (lane 6 of III, VI, IX, and XII in Supplementary Fig. 6). However, the mismatched bases have a differential impact on the hydrolysis of substrates with gap combinations to produce ROX signals (especially the mismatched strands of B24 and B25 in Supplementary Fig. 11c). Therefore, the mismatched base is one of the possible reasons for the observation that the substrates with the gap region formed by B25 with B11 to B16 are less amenable to be hydrolyzed by Exo λ after the endonuclease hydrolysis of Exo III.

To further investigate the cooperative mechanism of the enzymes, hydrolysis tests of substrates without gap were conducted, as illustrated in Fig. 3a, b. The complexes formed by A and B1 were hydrolyzed to varying degrees (Fig. 3c), and the complexes of AB11 or AB12 could be hydrolyzed within 1 h (Supplementary Fig. 12b). Similarly, the generated fluorescence from the opened F1 (Supplementary Fig. 12c) showed similar attenuation as observed in Supplementary Fig. 8. All the complexes of AB2 were unable to be completely hydrolyzed within 1 h (Supplementary Fig. 13b). Moreover, the hydrolyzed products were also difficult to observe in the case of AB23 to AB26 (Supplementary Fig. 13b). Therefore, B1 (B2) had a good performance on promoting (inhibiting) the endonuclease process of Exo III at the AP site. Interestingly, the endonuclease rate of Exo III decreased with the variation of AB11 to AB15 (blue columns in Fig. 3c, lanes 5 and 8 in Supplementary Fig. 12b) and increased in the case of AB16 (lane 8 in Supplementary Fig. 12b). The variation of the hydrolysis rate may be attributed to the increase of the exposed bases in A during the variation of AB11 to AB16, leading to the unexpected secondary structures formed in the dangling region of A (Supplementary Fig. 14), which triggers the endonuclease activity of Exo III at the AP site. The hydrolysis of AB16B2 without the dangling region (lanes 5 and 8 of XVI, XVII, and XVIII in Supplementary Fig. 6) had a lower rate compared to the hydrolysis of AB16 with the dangling region (lane 8 in Supplementary Fig. 12b). Thus, the conformational change of the secondary structure formed by the dangling end of A is one of the possible causes triggering the endonuclease process of Exo III. However, AB16 can be cleaved under the endonuclease activity of Exo III to produce A1B16, which is scarcely hydrolyzed by Exo III to generate B1 (Supplementary Fig. 12b) or trigger fluorescence signals (Supplementary Fig. 12c). This is because the shortening length of B1 leads to an increase in the 3' overhang of A1, resulting in a decrease in the exonuclease hydrolysis of Exo III (yellow column of AB16 in Fig. 3c, lane 9 in A1B16 of Supplementary Fig. 15b), which is an effective method to inhibit the leakage from the unexpected hydrolysis of Exo III[40,41]. The fluorescence of mismatched AB2 is shown in Supplementary Fig. 16.

**Signal transmission strategy based on gap-regulated exonuclease hydrolysis**

Based on the investigation conducted on the hydrolysis of complexes with or without gap, the complexes without gap in Fig. 3a effectively promotes the hydrolysis process (columns of AB11, AB12, and AB13 in Fig. 3c), whereas the complexes in Fig. 3b demonstrates a good inhibitory capability for hydrolysis (columns of AB24, AB25, and AB26 in Fig. 3d). Therefore, we performed hydrolysis tests on the complexes of AB24, with B1 as the input, to develop a signal transmission strategy with catalytic functionality, as

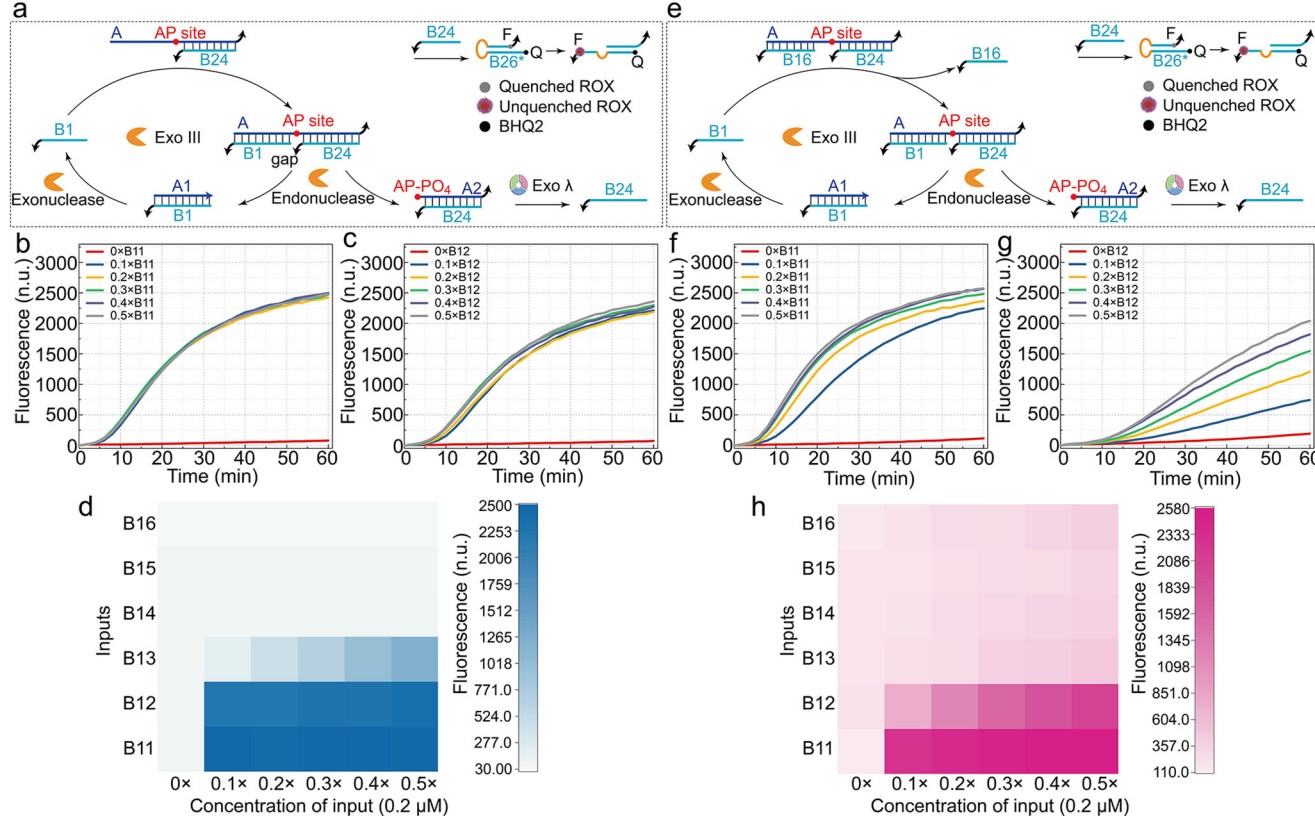

**Fig. 4 | Signal transmission strategy based on gap-regulated exonuclease hydrolysis. a** Schematic illustration of the signal transmission strategy without DSD. Fluorescence monitoring of the catalytic reaction with the input concentrations of B11 (**b**) and B12 (**c**) ranging from 0.1× to 0.5 × 0.2 μM. **d** Fluorescence influenced by the types and concentrations of the inputs. **e** Schematic illustration of the signal transmission strategy with DSD. Fluorescence monitoring of the catalytic reaction with the input concentrations of B11 (**f**) and B12 (**g**) ranging from 0.1× to 0.5 × 0.2 μM. **h** Fluorescence influenced by the types and concentrations of the inputs.

illustrated in Fig. 4a. Upon the addition of B1, Exo III cleaves AB1B24 at the AP site and proceeds to hydrolyze the remaining A1 in the 3′ direction, releasing the original input signal B1 as a catalytic reaction (Fig. 4b, c). A2B24 cleaved from Exo III with 5′ phosphate can be directly hydrolyzed by Exo λ, thus continuously releasing B24 as an amplified signal. The input tests were conducted on B11 to B16 (Fig. 4d). The concentration of B11 or B12 cannot affect the output rate, indicating that the concentration of B11 or B12 (0.1−0.5 × 0.2 μM) was excessive for the catalytic reaction (Fig. 4b, c). After reducing the input concentration to 0.01−0.05x (Supplementary Fig. 17e, f), the reaction rates showed differentiation, and the discrimination between positive and negative signals remained distinct, which suggests that the limit of the input concentration for the catalytic reaction is at least 2 nM (gray curve in Supplementary Fig. 17e), and the catalytic strategy has strong orthogonality between the input and output signals. The excessive A is consumed through hydrolysis, and the catalytic reaction does not produce any byproducts that may affect the reaction, apart from the input and output signals (Supplementary Fig. 18a, b). Furthermore, in a typical DSD reaction, an input with 3′ (5′) end requires displacement from the 5′ (3′) toeholds of another DNA strand to produce an output, and the ideal input-to-output signal ratio is 1:1. However, our strategy eliminates the polarity between the input and output signals by the cooperative hydrolysis. Moreover, the rate of signal production can be regulated by the length of the gap and the concentration of the input signal (Fig. 4b, c and Supplementary Fig. 17). Therefore, the catalytic strategy provides a solution for signal attenuation during long-distance signal transmission, and offers extensive possibilities for amplifying weak signals.

The exposed region of A in Fig. 4a may lead to the formation of complex secondary structures during the elaborate sequence design. Moreover, similar leakage as lanes AB16 with enzymes in Supplementary Fig. 12b may be produced in one system, where there are different kinds of

exposed regions with the similar structure of AB24. The sequence design of the exposed region increases the time and experimental costs. The pre-constructed gap region has good leakage suppression effect, especially the AB16B24 (Supplementary Fig. 19e). Therefore, B16 was used to protect the exposed region in AB24, which avoids the formation of secondary structures or extensive hybridization with similar structures. The pre-fabricated gap formed by B16 and B24 prevents the hydrolysis by Exo III (lanes 8 and 9 of XVII in Supplementary Fig. 6), as shown in Fig. 4e. The DSD reaction with the input of B1 is required to trigger the hydrolysis of Exo III (Fig. 4h), which is still a catalytic reaction (Fig. 4f, g and Supplementary Fig. 20). This method of protecting the dangling region reduces the possibility of secondary structures and simplifies the sequence design process, which allows for the coexistence of similar structures in a system without cross-interference, and expands the application space of the strategy.

## Solution to the MWCP

To demonstrate the compatibility and versatility of the signal transmission strategy mentioned above, a molecular computing model with a corresponding computational algorithm was established for solving the NP-hard problem of the MWCP. The MWCP, as a general form of the maximum clique problem, is widely used in protein structure prediction, computer vision, and robotics. Its objective is not to find the maximum clique in the graph but to search for a complete subgraph with the maximum weight. The normal form of the MWCP is described as follows. Consider an undirected graph $G(V, E, W)$, where $V$ is defined as the set of vertices; $E$ is the set of edges formed by the vertices in $V$; and $W$ represents the weights corresponding to each vertex. $\bar{G}(V, \bar{E}, W)$ is defined as the complement graph of $G(V, E, W)$, where $\bar{E}$ is the set of edges formed by the pairs of vertices in $V$ that are not connected: $\bar{E} = \{(v_i, v_j)|\forall(v_i, v_j) \notin E\}$. Every pair of vertices is connected as a subset $S$ of $V$, which is considered a clique in $G$, and $S$ is a

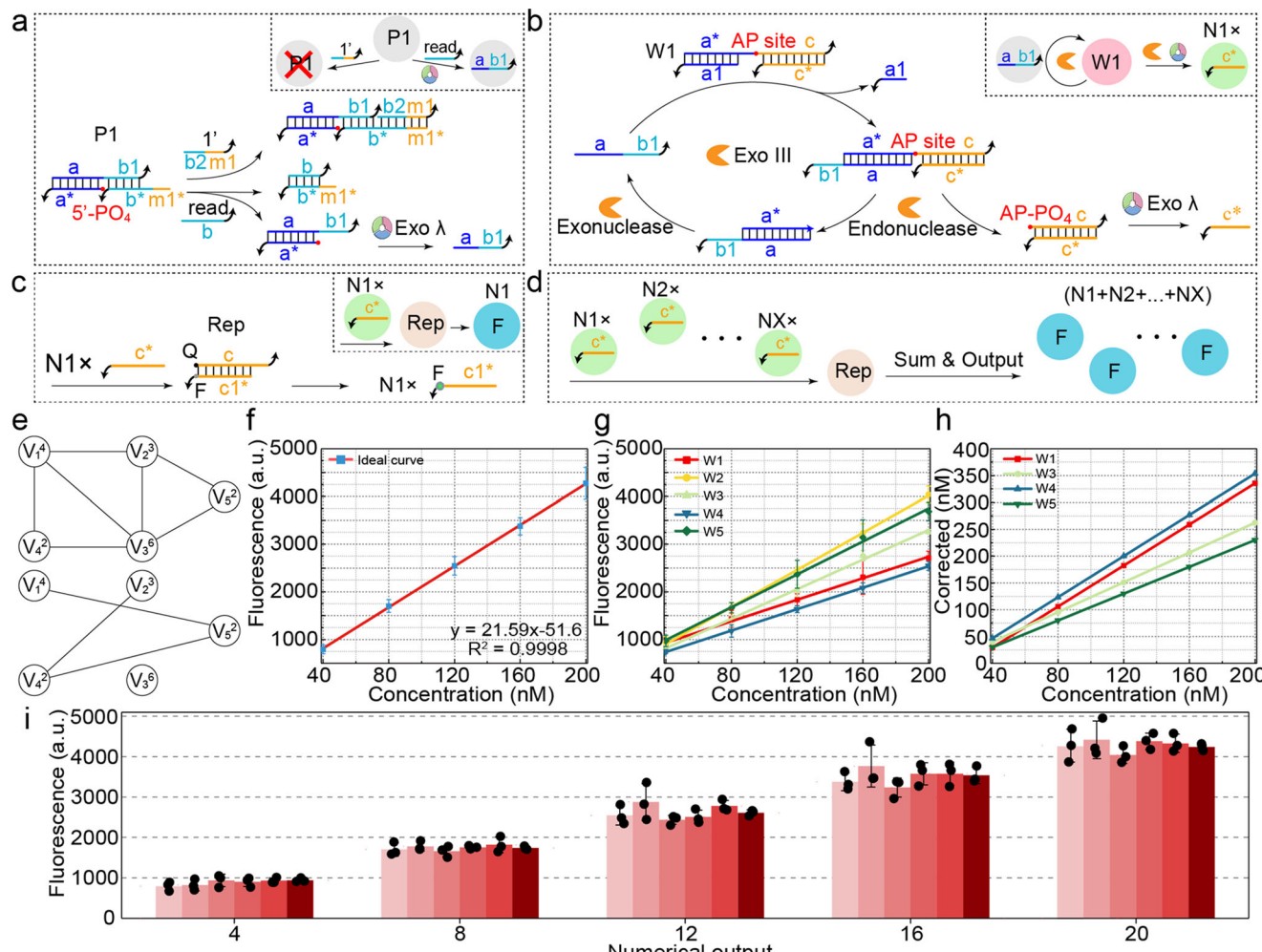

**Fig. 5 | Molecular computing modules and substrate correction for solving the MWCP.** Schematic illustration of the Point module (**a**), Weight module (**b**), and Sum & Output module (**c**) for vertex 1. **d** Schematic illustration of the Sum & Output module to sum all output strands in the form of fluorescence. **e** An undirected graph with five weighted vertices and its complement graph. The subscript and superscript of each vertex denote the vertex index and the vertex weight, respectively. **f** The ideal fluorescence response of the reporter triggered by the gradient inputs. **g** The actual fluorescence response of the gradient substrates in the Weight module. **h** The concentration correction of the substrates in the Weight module. **i** The fluorescence response of the corrected concentration of the substrates in the Weight module. The error bars represent the standard deviation of three sets of data ($n = 3$ independent experiments).

clique with the largest total weight of its vertices[42]:

$$W(S) = max \sum_{i=1}^{|V|} w_i v_i \tag{1}$$

$$v_i = \begin{cases} 1, v_i \in S \\ 0, v_i \notin S \end{cases} i = 1, 2, \ldots, |V| \tag{2}$$

$$v_i + v_j \leq 1, \forall \left(v_i, v_j\right) \in \bar{E} \tag{3}$$

Taking the example of the undirected graph shown in Fig. 5e, which is connected by five weighted vertices, a complement graph is defined as a graph that complements the given undirected graph. By connecting the disconnected vertices, the complement graph can be obtained, which possesses the following edges: (1, 5), (2, 4), and (4, 5). Here, the molecular computing model comprising a Point module, Weight module, and Sum & Output module was employed to solve the MWCP (Supplementary Fig. 21). All modules associated with vertex 1 as an example are illustrated in Fig. 5a–c.

The Point module corresponds to an individual vertex in the undirected graph and is used to screen all the vertices that need to be output in the

final clique. The Point module incorporates a previously proposed strategy in our work, which exhibits excellent leakage resistance and ensures the orthogonality of the input and output signals[43]. In Point1 of vertex 1 shown in Fig. 5a, the DNA duplex with a 5' recessed end is blocked by m1*b*, preventing the hydrolysis of Exo λ until m1*b* is removed. The signals of 1' and "read" are input to inhibit and activate Point1, and only the uninhibited vertices can receive "read" signals to output ab1 (Supplementary Fig. 22). Furthermore, all vertices share the same sequence b* for receiving the "read" signal, and the domain m1 of Point1 is unique to differentiate the inhibitory signals among all vertices. (Point module in Supplementary Fig. 21).

The Weight module represents the weights associated with each vertex and receives the output signals from the Point module, and performs the weighted calculations to amplify the signals based on the corresponding weights. The specific principle of Weight1 for vertex 1 is shown in Fig. 5b. Due to the incompleteness of biochemical reactions, the weight substrates may not be fully consumed to achieve preset weight multiplication. Therefore, the amount of the weight substrates needs to be adjusted to fit the ideal fluorescence curve by calculating the deviations between the ideal curve (Fig. 5f and Supplementary Fig. 23) and the practical curves (Fig. 5g and Supplementary Figs. 24–28). Then, all signals produced by the Weight module can be amplified in a predetermined magnification factor with the identical concentration of the input by correcting each weight substrate

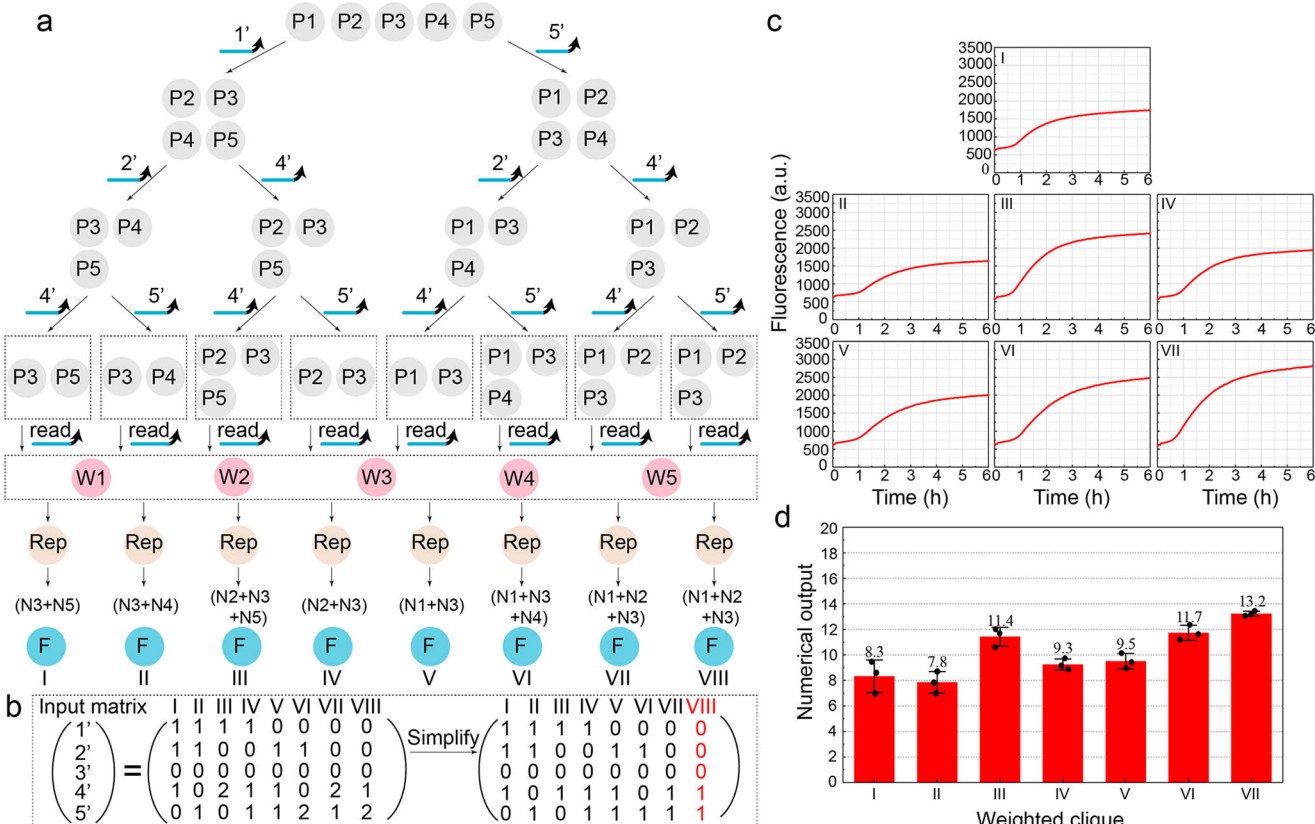

**Fig. 6 | Solution to the MWCP. a** Algorithm for the MWCP in the undirected graph with five vertex weights. **b** Input matrix for the MWCP. **c** Fluorescence result of each weighted clique. **d** Numerical output corresponding to the fluorescence value of each weighted clique. The error bars represent the standard deviation of three sets of data ($n = 3$ independent experiments).

(Fig. 5h), as shown in Fig. 5i. The detailed method of correcting practical curves to the ideal curve is shown in Supplementary Fig. 24. It was noticed that the concentration of W2 does not need to be corrected (Supplementary Fig. 25). Finally, the Sum & Output module is shared for all vertices and is used to receive the output signals from the Weight module in order to sum all the outputs in the form of fluorescence signals, as shown in Fig. 5c, d.

After establishing the molecular computing model, we designed a corresponding algorithm to search for the clique with the maximum weight, as shown in the tree in Fig. 6a. Since any edge in the complement graph cannot exist in one clique, meaning that two vertices connected by an edge cannot both be present, the set of all vertices in the undirected graph can be classified into two sets during the initial screening by the first edge (1, 5): The set that includes vertex 5 but not vertex 1, which is $(V_2^3, V_3^6, V_4^2, V_5^2)$; the set that includes vertex 1 but not vertex 3, which is $(V_1^4, V_3^3, V_5^6, V_2^2)$. A second round of screening is applied to these two sets, resulting in four different sets. The screening process continues iteratively until all edges of the complement graph have been traversed, thus searching for all cliques in the original undirected graph. In this example, there are eight cliques, labeled as I to VIII. The input to the complement graph can be represented as a vector $(1', 2', 3', 4', 5')^T$, as shown in the Input matrix in Fig. 6b. The excessive inputs beyond 1 to the matrix that have no contribution to the calculation may interfere with the true results. Therefore, the inputs greater than 1 can be replaced with 1, which causes the presence of identical vectors in the Input matrix (VIII, marked red in Fig. 6b). The matrix can be further simplified by removing the same vector, and the computations can be performed simultaneously in seven independent groups. During the continuous detection of fluorescence (Fig. 6c), the outputs of the seven groups basically reach stability. The corresponding numerical outputs are shown in Fig. 6d. The expression method based on fluorescence signals is consistently subject to the influence of uncertain factors within chemical reaction processes,

thereby impeding the precise identification of the ultimate numerical values. It is concluded that the MWC of the given graph is $(V_1^4, V_2^3, V_3^6)$ with the total weight of 13, which is ultimately manifested as 13.3 in the numerical output. The concurrent and intuitive output of all computational outcomes facilitates the feasibility of large-scale parallel computing.

## Discussion

We investigated the mechanism of the gap-regulated exonuclease hydrolysis and developed a signal transmission strategy capable of generating amplified DNA signals comparable to DSD. We demonstrated the potential of this strategy in solving the MWCP. During the hydrolysis of the substrate with gap combinations by two enzymes, the presence of an AP site is a crucial prerequisite for initiating Exo III cleavage and further triggering Exo λ exonuclease activity. The number of bases at the gap region could modulate the cooperative hydrolysis process, confirming that a fully double-stranded structure is not necessary for the endonuclease activity of Exo III, and partial dsDNA exposed at the AP site is sufficient. Moreover, two ssDNAs forming the gap domain opposite the AP site could respectively promote or inhibit the hydrolysis of Exo III. Thus, the cooperative hydrolysis process of these two enzymes can be controlled through DNA inputs, implementing a transmission strategy with orthogonal input-output signals. This strategy not only overcomes the polarity issue in conventional DSD and enables higher signal output yields, but also allows for signal detection and distinctive positive/negative signal contrast of DNA at a concentration as low as 2 nM.

The gap-regulated mechanism can be combined with DSD, where the substrate needs to be protected by a ssDNA, thus enabling the stable formation of a double-stranded structure from the exposed single-stranded domain that may otherwise form unknown secondary structures. Moreover, an artificially designed gap resistant to hydrolysis can be pre-constructed to

minimize leakage and crosstalk of the initial substrate even under the presence of exonucleases, which allows for more flexible sequence design and broader application possibilities to reduce time and experimental costs. Differences in scalability and sensitivity exist between the signal transmission strategies with and without DSD. Concerning sensitivity, the weaker signals can be recognized by the signal transmission strategy without DSD (Supplementary Fig. 17f compared to Fig. 4g). However, a system with excessive sensitivity may be susceptible to misinterpretation of irrelevant signals, leading to the leakage and crosstalk, thereby impeding the expansion of a large-scale system; The integration of DSD introduces pre-constructed gap structures, as illustrated in Supplementary Fig. 19, which confer advantages in reducing leakage. The less leakage is a necessary condition to meet the scalability requirements of the system, constituting a primary reason for selecting the signal transmission strategy with DSD to address the MWCP. Furthermore, the unified transmission method of signals is another important factor for the scalability of the system. It can be noted that the input and output of the signal transmission strategy are DNA strands with orthogonal relationships, which reduces crosstalk between signals and ultimately converts DNA signals into fluorescent signals with linear characteristics, meeting the quantitative analysis requirements of the results. Therefore, the signal transmission strategy with DSD has relatively good scalability in the system, and cascading between modules can be achieved.

However, the construction of molecular systems is inherently constrained by biochemical reactions. Due to the incompleteness of the reactions, adjustments to the concentrations of the substrate are required to achieve the desired output. In addition, despite performing rounding to achieve the final result, certain deviations persisted, which were particularly pronounced when employing two close but different fluorescence signals to represent adjacent integers, necessitating an exceptionally accurate control of the fluorescence output. Therefore, achieving ideal numerical output is a complex and challenging process. The study was intended to a proof of concept, and in future research, precise input-output ratios or near-perfect reaction efficiency may be achieved, thus promoting the optimization of DNA reaction systems.

The use of enzymes may bring some negative effects. Primarily, the specific reaction conditions are necessary for the system, including parameters such as the proportional composition of various components in the reaction solution. Additionally, maintaining the reaction temperature within the enzyme's operational range and regulating the reaction time in accordance with the enzyme's lifecycle are critical requirements. Furthermore, the use of enzymes requires that the design of substrates, inputs, and reports in the system meet specific requirements to prevent incorrect hydrolysis and signal leakage. The incorporation of enzymes may also introduce limitations to the application of complex DNA structures. Specifically, some enzymes exhibit diminished activity in acidic environments, making them unable to coexist with specialized DNA structures exclusively formed under acidic conditions, such as Hoogsteen triplexes and i-motifs. Moreover, large complex structures such as DNA tiles and origami may have sites for enzymatic hydrolysis, hindering the application of complex structures within enzyme-rich environments.

In conclusion, we experimentally demonstrated the hydrolytic mechanism of the gap-regulated enzymes and proposed a strategy in which the input signal catalyzes the generation of output signals with orthogonal characteristics. Furthermore, the combination of this strategy with DSD improved the performance in mitigating leakage and crosstalk, thus providing potential solutions for the NP-hard problem of the MWCP. The discovery of these mechanisms and strategies not only broadens our knowledge of these enzymes but also promises wide applications in signal detection, molecular computing, and nanomachines.

## Methods
### Native PAGE experiments
All the DNA sequences were provided in Supplementary Tables 1 and 2. The native PAGE experiments were performed at 80 V for 2.5 h by using the DYY-6D electrophoresis apparatus (Beijing Liuyi Co., Ltd., Beijing).

The freshly annealed substrates, exonucleases, and corresponding input DNA were added to a final volume of 20 μl 1× TAE/Mg$^{2+}$ buffer (pH = 8.0) for 1 h incubation at 30 °C, whereas the incubation for the experiment in Supplementary Fig. 10c was 0.5 h. Then, 60% glycerol (3 μl) was mixed with the incubated solution before running the 12% native polyacrylamide gel. Then, the gels were immediately moved into the Stains All solution for the 20 min dyeing operation after electrophoresis. Finally, the gels were destained by exposure to light (fading treatment) for 10–15 min to collect the clear images. For all gel electrophoresis experiments, with the exception of the input concentration gradient experiment in Supplementary Fig. 18, the substrates and the input concentration are 1 × 1 μM. Therefore, the standard concentration (1 × 1 μM) for the PAGE experiments was calculated as 20 pmol divided by 20 μl, and the input concentration gradient in Supplementary Fig. 18 is 0.1–0.5 × 1 μM. The detailed methods for the experiment were provided in Supplementary Methods.

### Fluorescence experiments
The fluorescence experiments were performed using a TECAN Microplate Reader Spark 20M (Tecan Trading Co., Grödig) at 30 °C (±0.5 °C) and were independently performed with three replications for the final fluorescence curves or columns to minimize fluctuations and prove the repeatability of the fluorescence experiments. The ex-em wavelengths used in the measurements were set as 480–525 nm for the FAM label and 560–606 nm for the ROX label. In the fluorescence experiments except for Fig. 6c, the freshly annealed substrates were mixed to 1 × TAE/Mg$^{2+}$/ BSA buffer (pH = 8.0) with a final volume of 100 μl, and the initial state without inputs was recorded as the baseline. Then, the kinetic cycles were paused to add the corresponding inputs after the first three cycles. All the fluorescence data was recorded at 1 min or 2 min interval. Similar to the concentration setting in the gel electrophoresis experiment, a standard concentration (1 × 0.2 μM) is set to regulate the experimental operation and data expression, which was calculated as 20 pmol divided by 100 μl. In the fluorescence figures without labeled concentrations (Figs. 1–3 and Supplementary Figs. 1–5, 7–9, 11–13 and 16), the concentration of all components is set as 1 × 0.2 μM. In the fluorescence figures with labeled concentrations (Fig. 4 and Supplementary Figs. 17 and 19), the input DNA signal is $n$ × 0.2 μM ($n$ represents the specific parameters in each Figure), and the concentrations of substrate and fluorescent reporter are 1 × 0.2 μM. The substrate concentration gradient in Fig. 5 and Supplementary Figs. 23–28 are data sampling points, and the input signal and fluorescence reporter concentrations are 0.2× and 1 × 0.2 μM, respectively. The input signal and fluorescence reporter concentration of Fig. 6 are 0.2× and 1 × 0.2 μM, respectively. The concentrations of W1-W5 as setting weights need to be converted according to the correction curve, which are 29.4, 30, 66.5, 8.4, 4.9 nM, respectively. For the fluorescent measurements in Fig. 6c, the Input matrix was added into seven tubes, and all freshly annealed substrates of the Point module were mixed for 2 h incubation. Then, the "read" signal, 5 U Exo III, and Exo λ were added to seven tubes for 1 h incubation. The freshly prepared substrate of the Sum & Output was added to seven tubes for the fluorescent measurement, and the first three data measurements in the kinetic cycles was recorded as the initial values. Then, the kinetic cycles were paused to add all freshly annealed substrates of the Weight module for the remaining measurements. The fluorescence data was recorded at 2 min intervals.

### Statistics and reproducibility
The details about proportion of each component, operation steps, parameters, and process of fluorescence data are provided in relevant methods sections, figure legends, and tables where applicable.

### Reporting summary
Further information on research design is available in the Nature Portfolio Reporting Summary linked to this article.

## Data availability

The source data for all figures of fluorescence is available in Supplementary Data 1 and 2. All figures are open on figshare (https://figshare.com/s/3c62963ac3e5ee2521c8). Uncropped gels are included in Supplementary Figs. 29–37. Any further data not included in the text will be made available upon request.

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

## Acknowledgements

This work is supported by 111 Project (No. D23006), the National Natural Science Foundation of China (Nos. 62272079, 61972266), Natural Science Foundation of Liaoning Province (2022-KF-12-14), the Postgraduate Education Reform Project of Liaoning Province (No. LNYJG2022493), the Artificial Intelligence Innovation Development Plan Project of Liaoning Province (No. 2023JH26/10300025), the Dalian Outstanding Young Science and Technology Talent Support Program (No. 2022RJ08), Dalian Major Projects of Basic Research (No. 2023JJ11CG002).

## Author contributions

Q.Z. directed and funded this research. X.L. found the mechanism and strategy. X.L., X.Z., and S.C. designed and performed the experiments. X.L., X.Z., S.C., and S.X. conceived the research. X.L., X.Z., S.C., and S.X. analyzed the data and wrote the initial draft of the manuscript. R.L., B.W., X.W., and Q.Z. reviewed the manuscript and supervised this research. All authors have discussed and checked the results, and agreed to the final version of the manuscript.

## Competing interests

The authors declare no competing interests.
