## [Peer Review File · Communications Biology]

Reviewers' comments:

Reviewer #1 (Remarks to the Author):

This paper presents a novel signal transmission strategy that has the potential for a broader scope of applications in DNA computing.

Here are some detailed comments:

(1) The authors need to add more details about the scalability of the strategy in the discussion section.

(2) It would be helpful if the leakage reduction was analyzed more quantitatively.

(3) Using enzymes can have some side effects, so please discuss it in detail.

Reviewer #2 (Remarks to the Author):

In this manuscript, Liu et al. report a novel strategy to overcome signal attenuation in DNA strand displacement reactions due to the accumulation of displaced DNA strands by combining the hydrolytic activities of Exo III and Exo- λ .

The apurinic/apyrimidinic endonuclease and subsequent creation of a gap at the AP site by the 3' exonuclease activity of the Exo-III family of enzymes are well known in the context of base excision repair. The authors have coupled this process with the 5' exonuclease activity of Exo- λ to devise a signal transmission method with DNA strands as inputs and turn-on fluorescence signals as output. They identified the optimal length of the gap at the AP site of the DNA device and showed how this strategy works in a catalytic strand displacement cycle. Finally, they implemented this strategy to solve the maximum weight clique problem using DNA sequences. This work is likely to be of interest in the field of DNA nanotechnology and molecular computation. The approach reported here will help improve the design of DNA devices and dynamic DNA-based biochemical networks that employ strand displacement reactions.

The experiments were designed and executed well, and I recommend publishing this article in Communications Biology. However, there is a lack of clarity in how the experiments were performed and the manuscript can benefit from proofreading.

Minor points:

1. The data shown in Fig. 1 and 2 suggest that the presence of Exo- λ could decrease the fluorescence signal from F1.

a. In Fig. 1b, the FAM fluorescence intensity at 60 min was higher for AB+F1+F2+III (curve no. 2) than for AB+F1+F2+III+ λ (curve 4 in the same plot). Why does the FAM (F1) fluorescence signal decrease with the addition of Exo- λ ?

b. In Fig. 3c, why does the addition of Exo- λ decrease FAM fluorescence in the case of AB11?

2. Typically, 2.5 mM Mg²⁺ is used for optimal 5' exonuclease activity of Exo- λ . Is the enzyme activity affected by 12.5 mM Mg²⁺ in the buffer?

3. What is the "fading treatment" the gels are subjected to before imaging?

4. In the methods section, the concentration of substrate strands, F1, and F2 must be mentioned to help others reproduce the experiments. This information is not apparent from the sentence beginning with "The standard concentration (1 × 0.2 μ M)...".

5. The reaction buffer contains 50 $\mu\text{g/ml}$ BSA. The purpose of BSA in the reaction is not clear.
6. Consider rephrasing "Cooperative hydrolysis of Exo III and Exo λ ." It is misleading and makes one think that the Exo-III and Exo- λ are getting hydrolyzed.

Point-by-point reply:

We highly appreciated the reviewers' comments, whom we thank for their constructive criticism. We addressed the raised points as follows:

Reviewers' comments

Reviewer #1 (Remarks to the Author)

This paper presents a novel signal transmission strategy that has the potential for a broader scope of applications in DNA computing.

Reply:

We thank the reviewer for the positive words and valuable constructive suggestions on our study.

(1) The authors need to add more details about the scalability of the strategy in the discussion section.

Reply:

We appreciate the reviewer's suggestion. We have added the relate content to illustrate the scalability of the proposed strategy in the discussion section (lines 391-407 of Article). The detailed content is as follows: Differences in scalability and sensitivity exist between the signal transmission strategies with and without DSD. Concerning sensitivity, the weaker signals can be recognized by the signal transmission strategy without DSD (Supplementary Fig. 17f compared to Fig. 4g). However, a system with excessive sensitivity may be susceptible to misinterpretation of irrelevant signals, leading to the leakage and crosstalk, thereby impeding the expansion of a large-scale system; The integration of DSD introduces pre-constructed gap structures, as illustrated in Supplementary Fig. 20, which confer advantages in reducing leakage. The less leakage is a necessary condition to meet the scalability requirements of the system, constituting a primary reason for selecting the signal transmission strategy with DSD to address the MWCP. Furthermore, the unified transmission method of signals is another important factor for the scalability of the system. It can be noted that the input and output of the signal transmission strategy are DNA strands with orthogonal relationships, which reduces crosstalk between signals and ultimately converts DNA signals into fluorescent signals with linear characteristics, meeting the quantitative analysis requirements of the results. Therefore, the signal transmission strategy with DSD has relatively good scalability in the system, and cascading between modules can be achieved.

(2) It would be helpful if the leakage reduction was analyzed more quantitatively.

Reply:

We thank the reviewer for pointing out this important issue. The leakage we proposed occurred in lanes AB16 with enzymes in Supplementary Fig. 12b, which is difficult to

quantitatively analyze through fluorescence signals due to the inability to produce B16. Therefore, the software Image J is applied to quantitatively analyze the grayscale value of the gel and obtain the ratio of unhydrolyzed AB16 with the addition of enzymes, thereby calculating the proportion of leakage (Supplementary Figs. 20a and 20b as shown below). This method may have bias, but some authoritative works have applied this method for the quantitative analysis, which has relatively reliable credibility.^[1] The pre-constructed gap structure can inhibit the hydrolysis of AB16B2 to reduce leakage. B16 or B2, which triggers fluorescence signals, are difficult to be produced in XVI-XVIII of Supplementary Fig. 6. Therefore, the software Image J is also applied to quantitatively analyze the leakage (Supplementary Figs. 20c and 20d). Finally, the leakage of AB16 and the leakage of all AB16B2 structures were compared, and a relatively quantitative analysis of leakage reduction is obtained (Supplementary Fig. 20e). The added content can be found in lines 283-285 of Article and lines 203-218 of Supplementary Information.

Supplementary Figure 20. Quantitative analysis of leakage using software of Image J. **a**, The analysis of unhydrolyzed AB16 in Supplementary Fig. 12c by software Image J. The grayscale value of lane 7 is measured as standard value by software Image J, then, the ratio of unhydrolyzed AB16 in lanes 8 and 9 are calculated as the measured grayscale value of lanes 8 and 9 divided by the grayscale value of lane 7, respectively. **b**, The leakage rate of AB16 with enzymes in Supplementary Fig. 20a. The rate of hydrolyzed AB16 is calculated as 1 minus the rate of unhydrolyzed AB16, which is the leakage rate of AB16. **c**, The analysis of unhydrolyzed AB16B2 with enzymes in XVI-XVIII of Supplementary Fig. 6 by software Image J. The grayscale value of AB16B2 without enzymes is measured as standard value by software Image J. The ratio of unhydrolyzed AB16B2 with enzymes is

calculated as the measured grayscale value of AB16B2 with enzymes divided by the grayscale value of AB16B2 without enzymes. **d**, The leakage rate of AB16B2 with enzymes in XVI-XVIII of Supplementary Fig. 6. The rate of hydrolyzed AB16B2 with enzymes is calculated as 1 minus the rate of unhydrolyzed AB16B2 with enzymes, which is the leakage rate of AB16B2. **e**, The leakage reduction ratio of AB16B2 relative to AB16.

(3) Using enzymes can have some side effects, so please discuss it in detail.

Reply:

We appreciate the reviewer for raising these problems. As the reviewer pointed out, the introduction of enzymes into the system may bring some side effects, and we have added detailed content to illustrate these aspects in the Discussion section, which can be found in lines 418-430 of Article. The added content is as follows: The use of enzymes may bring some negative effects. Primarily, the specific reaction conditions are necessary for the system, including parameters such as the proportional composition of various components in the reaction solution. Additionally, maintaining the reaction temperature within the enzyme's operational range and regulating the reaction time in accordance with the enzyme's lifecycle are critical prerequisites. Furthermore, the use of enzymes requires that the design of substrates, inputs, and reports in the system meet specific requirements to prevent incorrect hydrolysis and signal leakage. The incorporation of enzymes may also introduce limitations to the application of complex DNA structures. Specifically, some enzymes exhibit diminished activity in acidic environments, making them unable to coexist with specialized DNA structures exclusively formed under acidic conditions, such as Hoogsteen triplexes and i-motifs. Moreover, large complex structures such as DNA tiles and origami may have sites for enzymatic hydrolysis, hindering the application of complex structures within enzyme-rich environments.

Reviewer #2 (Remarks to the Author)

In this manuscript, Liu et al. report a novel strategy to overcome signal attenuation in DNA strand displacement reactions due to the accumulation of displaced DNA strands by combining the hydrolytic activities of Exo III and Exo- λ .

The apurinic/aprimidinic endonuclease and subsequent creation of a gap at the AP site by the 3' exonuclease activity of the Exo-III family of enzymes are well known in the context of base excision repair. The authors have coupled this process with the 5' exonuclease activity of Exo- λ to devise a signal transmission method with DNA strands as inputs and turn-on fluorescence signals as output. They identified the optimal length of the gap at the AP site of the DNA device and showed how this strategy works in a catalytic strand displacement cycle. Finally, they implemented this strategy to solve the maximum weight clique problem using DNA sequences. This work is likely to be of interest in the field of DNA nanotechnology and molecular computation. The approach reported here will help improve the design of DNA devices and dynamic DNA-based biochemical networks that

employ strand displacement reactions.

The experiments were designed and executed well, and I recommend publishing this article in *Communications Biology*. However, there is a lack of clarity in how the experiments were performed and the manuscript can benefit from proofreading.

Reply:

We thank the reviewer for the positive comments and valuable constructive suggestions on our study.

Minor points:

1. The data shown in Fig. 1 and 2 suggest that the presence of Exo- λ could decrease the fluorescence signal from F1.

a. In Fig. 1b, the FAM fluorescence intensity at 60 min was higher for AB+F1+F2+III (curve no. 2) than for AB+F1+F2+III+ λ (curve 4 in the same plot). Why does the FAM (F1) fluorescence signal decrease with the addition of Exo- λ ?

b. In Fig. 3c, why does the addition of Exo- λ decrease FAM fluorescence in the case of AB11?

Reply:

We appreciate the reviewer for raising these issues. Overall, as the reviewer pointed out, the apparent cause for the decrease in FAM fluorescence signal in Figs. 1 and 2 is attributed to the addition of Exo- λ , but the mechanism of signal reduction differs between the two cases. From the perspective of fluorescence signal generation, the FAM signal in curve 4 of Fig. 1b can be attributed to the opening of F1 facilitated by either A2B or B. However, the FAM signals observed in Figs. 2 and 3 are exclusively generated by the action of B1 in opening F1. It is noteworthy that B1 constitutes a partial sequence of B, whereas B has the capability to concurrently open both F1 and F2. This distinction serves as the underlying reason for the divergent mechanisms leading to the reduction in FAM signals. The detailed analysis follows:

a. In response to the queries, we have delved into more compelling factors concerning the FAM decrease in curve 4 of Fig. 1b. It is evident from the curve 8 in Fig. 1b that A2B operates normally in the process of opening F1 hairpins. The observed decrease in FAM signal is likely attributed to the process of B opening F1. The addition of Exo- λ directly promotes the generation of complete B. Consequently, we conducted an investigation into the fluorescence characteristics of F1 and F2 opened by complete B, as illustrated in the Supplementary Fig. 1 (as shown below). It can be observed that B is not ideal for triggering both F1 (curve 2 in Supplementary Fig. 1c) and F2 (curve 2 in Supplementary Fig. 1d) simultaneously, particularly with a more pronounced impact on the FAM signal than ROX signal. The isolated opening of F1 by B remains unaffected, while a decrease in FAM fluorescence occurs with the addition of F2 (curve 3 in Supplementary Fig. 1c), then, ultimately reducing to nearly the same fluorescence level as the simultaneous opening of F1 and F2 by B (curves 2 and 3 in Supplementary Fig. 1c). For this phenomenon, we

experimentally verified the structure of B when simultaneously opening F1 and F2 (lane 6 in Supplementary Fig. 1b), which confirmed that the complete B can form a stable structure with F1 and F2.

In the structural simulation depicted in Figure a for BF1F2, the physical distance between BHQ2 and FAM modification in BF1F2 is relatively close. The ex-em wavelengths used for FAM detection in fluorescence experiments are set to 480-525 nm, and the quenching range of BHQ2 is 550-650 nm, with a distance of 25 nm from the predetermined emission wavelength of FAM fluorescence. The quenching wavelength of the quencher does not strictly meet the wavelength range mentioned above, but conforms to a normal distribution primarily concentrated between 550-650 nm. Consequently, BHQ2 exhibits quenching capability for fluorescence at the 525 nm wavelength, leading to partial quenching of the excited FAM fluorescence. Finally, in curve 4 of Fig. 1b, the addition of Exo- λ promotes the generation of B, ultimately contributing to the formation of the BF1F2 structure. However, BHQ2 in BF1F2 has a certain quenching effect on FAM fluorescence, resulting in the fluorescence intensity of curve 4 being lower than that of curve 2 in Fig. 1b. The detailed content has been added to lines 153-157 of Article and lines 75-79 of Supplementary Information.

Supplementary Figure 1. The analysis of FAM fluorescence decrease. a, The reaction and structural simulation of B, F1, and F2. **b**, The PAGE analysis of B opening F1 and F2. **c**, FAM fluorescence testing of B opening F1 and F2. **d**, ROX fluorescence testing of B opening F1 and F2.

b. The reason for the fluorescence decrease in Fig. 3c is consistent with that in Fig. 2, because the fluorescence signals in both figures are triggered by the same product B11. We have explained the specific details in lines 198-203 of Article, which is as follows: The 5' end of B1 with a dangling tail in the opened F1 (Fig. 2a) is a potential substrate for Exo λ hydrolysis.^[2] Therefore, the impact of different enzymes added during the opening of the hairpin F1 on fluorescence signals was tested. We found that the addition of Exo λ is the primary cause of signal attenuation, particularly in the cases of B11 and B13 (Supplementary Fig. 8). Moreover, the continuous reduction in the length of the 5' end of B1 promotes the opened F1 to close, resulting in a decrease in fluorescence signal.

2. Typically, 2.5 mM Mg²⁺ is used for optimal 5' exonuclease activity of Exo-λ. Is the enzyme activity affected by 12.5 mM Mg²⁺ in the buffer?

Reply:

We appreciate the reviewer's concern. As pointed out by the reviewer, 2.5 mM Mg²⁺ is used for optimal 5' exonuclease activity of Exo λ, and the influence of the concentration of Mg²⁺ has been explored in previous work.^[3] It confirms that Mg²⁺ above 2.5 mM in the buffer is not obvious to increase the hydrolysis rate of Exo λ, but the increase in the concentration of Mg²⁺ facilitates hybridization reactions between DNA strands, which is beneficial for DNA based reaction systems. In addition, the relevant work ^[4] has used a Mg²⁺ concentration of 12.5 mM, confirming that this concentration of Mg²⁺ is acceptable. Undoubtedly, orchestrating the concerted efforts of various working components within a relatively intricate reaction system poses challenges. Ensuring the optimal conditions for one factor may introduce bias to the whole system. Identifying the optimum environmental conditions is difficult in a large-scale system, although it is not the primary focus of this work. In the initial exploratory investigations, it is determined that the ion conditions at this concentration suffice to fulfill the requisites of the proposed system. Consequently, the impact of magnesium ion concentration on our system has not been experimentally assessed in this study.

3. What is the "fading treatment" the gels are subjected to before imaging?

Reply:

We thank the reviewer for raising this issue. After the electrophoresis of the gel is completed, DNA substrates without fluorescence modifications are generally not observable through visible or invisible light (figure of Unstained gel as shown below). Therefore, the gel is stained to facilitate a clearly observation and comparison of DNA bands within the gel. The staining solution consists of a 1:1 volume mixture of formamide and deionized water, followed by the addition of stains all powder (0.1 g/L). The staining solution exhibits stronger staining ability for DNA strands compared to the background in the gel (figure of Stained gel without fading treatment), but the stained gel without fading treatment is unfavorable for observation and imaging. Due to the characteristic of the staining solution to visible light decomposition, the stained gel will be subjected to a uniform fluorescent light source for a specified duration of fading treatment. The fading treatment continues until the DNA bands are distinctly visible, and the background of the gel approaches a nearly colorless state (figure of Stained gel with fading treatment), which ensures the clear experimental results. The gel figure as shown below serves merely as a demonstration of the staining and fading treatment, unrelated to the experiments in this article.

4. In the methods section, the concentration of substrate strands, F1, and F2 must be mentioned to help others reproduce the experiments. This information is not apparent from the sentence beginning with “The standard concentration ($1 \times 0.2 \mu\text{M}$)...”.

Reply:

We thank the reviewer for pointing out this issue. We have revised the unclear description of the experiment and added details on the concentration of each component to assist interested researchers in reproducing our experimental results. The specific content is as follows:

In the Methods - Native PAGE experiments section (lines 559-564 of Article): For all gel electrophoresis experiments, with the exception of the input concentration gradient experiment in Supplementary Fig. 18, the substrates and the input concentration are $1 \times 1 \mu\text{M}$. Therefore, the standard concentration ($1 \times 1 \mu\text{M}$) for the PAGE experiments was calculated as 20 pmol divided by $20 \mu\text{L}$, and the input concentration gradient in Supplementary Fig. 18 is $0.1\text{-}0.5 \times 1 \mu\text{M}$.

In the Methods - Fluorescence experiments section (lines 576-589 of Article): Similar to the concentration setting in the gel electrophoresis experiment, a standard concentration ($1 \times 0.2 \mu\text{M}$) is set to regulate the experimental operation and data expression, which was calculated as 20 pmol divided by $100 \mu\text{L}$. In the fluorescence figures without labeled concentrations (Figs. 1-3, Supplementary Figs. 1-5, 7-9, 11-13, and 16), the concentration of all components is set as $1 \times 0.2 \mu\text{M}$. In the fluorescence figures with labeled concentrations (Fig. 4, Supplementary Figs. 17 and 19), the input DNA signal is $n \times 0.2 \mu\text{M}$ (n represents the specific parameters in each figure), and the concentrations of substrate and fluorescent reporter are $1 \times 0.2 \mu\text{M}$. The substrate concentration gradient in Fig. 5 and Supplementary Figs. 23-28 are data sampling points, and the input signal and fluorescence reporter concentrations are $0.2 \times$ and $1 \times 0.2 \mu\text{M}$, respectively. The input signal and fluorescence reporter concentration of Fig. 6 are $0.2 \times$ and $1 \times 0.2 \mu\text{M}$, respectively. The concentrations of W1-W5 as setting weights need to be converted according to the correction curve, which are 29.4 nM , 30 nM , 66.5 nM , 8.4 nM , 4.9 nM , respectively.

5. The reaction buffer contains $50 \mu\text{g/ml}$ BSA. The purpose of BSA in the reaction is not clear.

Reply:

We thank the reviewer for pointing out this important issue. Bovine serum albumin (BSA)

is commonly employed as a stabilizing agent in the preservation and reaction solutions of enzymes. For most DNA substrates, BSA enhances the completeness of enzyme cleavage reactions, facilitating repeated cleavages. Furthermore, BSA contributes to the enhanced stability of the enzyme during a long period of cleavage reaction, and demonstrates the capacity to sequester metal ions and other chemical substances that may otherwise inhibit the activity of enzymes. In related experiments utilizing exonucleases, BSA is one of the components in the buffer solution, the concentration of which is typically selected as 50 or 100 $\mu\text{g/mL}$, [4, 5, 6] with the flexibility for adjustment based on specific requirements. It is noteworthy that an excessive concentration of BSA induces solution viscosity, which leads to the generation of numerous small bubbles during the oscillation process, thereby hindering the fluorescence measurement experiments. Consequently, a BSA concentration of 50 $\mu\text{g/mL}$ is employed in this work.

6. Consider rephrasing “Cooperative hydrolysis of Exo III and Exo λ .” It is misleading and makes one think that the Exo-III and Exo- λ are getting hydrolyzed.

Reply:

We appreciate the reviewer for raising this issue. We have replaced “Cooperative hydrolysis of ...” with “Cooperative hydrolysis driven by ...”, which can be found in lines 34-35, 114-115, 139, 142, 183-184 and 628 in Article, and lines 152-153 in Supplementary Information.

References

1. Zhang, C. et al. Programmable allosteric DNA regulations for molecular networks and nanomachines. *Sci. Adv.* **8**, pii: eabl4589 (2022).
2. Wu, T. B. et al. Noncanonical substrate preference of lambda exonuclease for 5'-nonphosphate-ended dsDNA and a mismatch-induced acceleration effect on the enzymatic reaction. *Nucleic Acids Res.* **46**, 3119-3129 (2018).
3. Hwang, W. et al. Dynamic coordination of two-metal-ions orchestrates λ -exonuclease catalysis. *Nat. Commun.* **9**, 4404 (2018).
4. Qin, Z. H., Liu, Y., Zhang, L. H., Liu, J. J. & Su, X. Programming dissipation systems by DNA timer for temporally regulating enzyme catalysis and nanostructure assembly. *ACS Nano* **16**, 14274-14283 (2022).
5. Yoo, J. & Lee, G. Allosteric ring assembly and chemo-mechanical melting by the interaction between 5'-phosphate and lambda exonuclease. *Nucleic Acids Res.* **43**, 10861–10869 (2015).
6. Wu, T. B. et al. DNA terminal structure-mediated enzymatic reaction for ultra-sensitive discrimination of single nucleotide variations in circulating cell-free DNA. *Nucleic Acids Res.* **46**, e24 (2018).

REVIEWERS' COMMENTS:

Reviewer #3 (Remarks to the Author):

The authors have responded to all the comments satisfactorily.

Please mention in the Methods section how long the gels were destained by exposure to light (fading treatment).

Point-by-point reply for the final revision:

Reviewers' comments

Reviewer #3 (Remarks to the Author)

The authors have responded to all the comments satisfactorily.

Please mention in the Methods section how long the gels were destained by exposure to light (fading treatment).

Reply:

We thank the reviewer for pointing out this important issue. The time parameter required for the fading treatment has been added to the Methods section. The added content is as follows: Finally, the gels were destained by exposure to light (fading treatment) for 10-15 min to collect the clear images.